

# Seismic section image detail enhancement method based on wavelet transform

Xiang-Yu Jia[1], Chang-Lei DongYe[1,2]

[1]College of Computer Science and Engineering, Shandong University of Science and Technology, Qingdao, 266590,
China;
[2]Shandong Province Key Laboratory of Wisdom Mine Information Technology, Shandong University of Science and
Technology, Qingdao, 266590, China;

*Correspondence to*: Chang-Lei DongYe (dycl.cn@ 163.com)

**Abstract.** The seismic section image contains a wealth of texture detail information, which is important for the
interpretation of the formation profile information. In order to enhance the texture detail of the image while keeping the
structural information of the image intact, a multi-scale enhancement method based on wavelet transform is proposed.
First, the image is wavelet decomposed to obtain a low frequency structural component and a series of high frequency
texture detail components; Secondly, bilateral texture filtering is performed on the low-frequency structural components
to filter out high-frequency noise while maintaining the edges of the image; adaptive enhancement is performed on the
high-frequency detail components to filter out low-frequency noise while enhancing detail; Finally, the processed high
and low frequency components are reconstructed by wavelet can obtained the seismic section image with enhanced
detail. The method of this paper enhances the texture detail information in the image while preserving the edge of the
image.

## 1. Introduction

Seismic section images have obvious texture features, and different texture regions represent different geological
bodies. It is of great significance for the interpretation of geological faults and other stratigraphic section information.
However, in the process of generating seismic section images, the texture of seismic section image is blurred or even
interrupted due to the influence of external noise on the collected seismic data, which causes interference to the
interpretation of the later seismic image. Therefore, studying the texture detail enhancement method of seismic section
image has both theoretical significance and potential application value.

At present, image detail enhancement methods can be divided into three categories: The first type is the airspace
domain enhancement method, and representative airspace enhancement methods include global histogram equalization
(Li et al., 2008; Pandey et al., 2017), local histogram equalization (Zhu et al., 1999; Stark, 2002; Cheng et al., 2004;
Hum et al., 2015), etc.; The former is due to the enhancement of the global image, does not consider the frequency and
detail information of the image, which is likely to cause excessive enhancement and cannot highlight the target
information in the image; The latter overcomes the defects of the foregoing method to a certain extent, but it only
considers the gray distribution in the local window and does not consider the characteristics of the overall image, which
tends to weaken the hierarchical sense of the image. The second type of method is a frequency domain enhancement
method, including homomorphic filtering (Voicu, 1997; Seow and Asari, 2006), high-pass filtering (Makandar and
Halalli, 2015; Balovsyak and Odaiska, 2018). This type of method is based on an illumination-reflection model that





transforms the multiplicative illumination and reflection components into an additive domain in the logarithmic domain. Then using a high-pass filter in the Fourier transform domain to enhance the high-frequency reflection component and suppress the low-frequency illumination component; The precondition for this type of method is that the image illumination is uniform, so the enhancement effect is poor for seismic section images with high and dark areas; In

addition, if the cutoff frequency of the high-pass filter is too high, it will result in severe compression of the dynamic range and loss of detail; Conversely, the dynamic range compression decreases. The third type is an image enhancement method based on transform domain. Such methods perform multi-scale decomposition of images by existing multi-scale transforms, such as wavelet transform (Tao et al., 2015; Makinana et al., 2016; Witwit et al., 2017) and Curvelet transform (Bhutada et al., 2011; Hashemahmed et al., 2015), and then stretch the transform coefficients, finally inverse

transform to obtain enhanced images. This multi-scale decomposition can effectively extract the feature information of the image, such as curves and textures. However, these multi-scale decompositions cannot avoid the occurrence of a circular effect at the edges of the image because they use a linear filter such as a Gaussian filter in the decomposition process. Therefore, for the seismic section image, which contains rich texture and curve information, it is necessary to construct multi-scale decomposition based on nonlinear edge-preserving filtering and apply it to seismic section image

processing. We propose a wavelet decomposition method based on bilateral texture filtering and texture detail adaptive enhancement. The method firstly performs wavelet decomposition on the image, performs bilateral texture filtering on the decomposed low-frequency components, and performs adaptive detail enhancement on the high-frequency components. The processed high and low frequency components are wavelet reconstructed to obtain a seismic section image with enhanced texture details. The biggest feature of this method is the ability to separate the texture details in the

image, while maintaining the edge information of the original image during processing. This property is very beneficial to the enhancement of image texture details.

In section 1 of this paper, the background and significance of the research are discussed. The deficiencies of the existing algorithms are analyzed for the characteristics of seismic profiles. The idea of image texture detail enhancement based on wavelet transform is proposed. Section 2 presents the seismic detail image texture detail enhancement model

and algorithm flow; Section 3 carries out image enhancement contrast experiment. The experimental results show that the proposed method achieves better enhancement effects on seismic section images with rich texture information, better embodying spatial details and texture features, and provides geoscientists with More informative information.

## 2. Model and algorithm

In this paper, the wavelet transform can effectively separate the characteristics of low frequency and high

frequency components of the image, and the seismic section image is decomposed into a low frequency structural image and multiple high frequency texture images by wavelet transform. On the low-frequency structural image, the bilateral texture filtering method is used to quickly and accurately estimate and remove the high-frequency noise of the image; On the high-frequency texture image, multi-scale detail enhancement and denoising are performed on the detail component reflecting the texture content of the image. Then, an adaptive optimization strategy is used to improve the

contrast of the image after the above processing to obtain the final enhanced image. The processing flow is shown in Fig. 1.


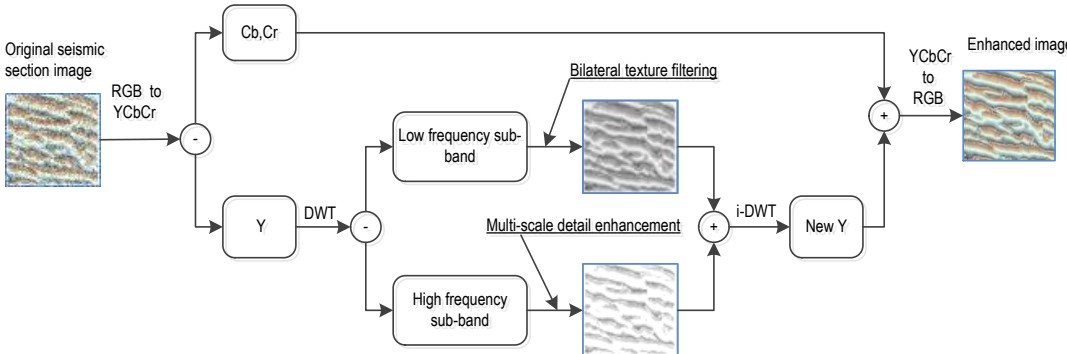

**Fig. 1 Process of seismic section image detail enhancement**

The image texture detail enhancement method shown in Figure 1 is as follows:

Step 1: Converting the seismic section image from the RGB space to the YCbCr space, and decomposing to obtain the luminance component $Y$, the chrominance components $Cb$ and $Cr$;

Step 2: performing single-layer discrete wavelet decomposition on the luminance component $Y$ to obtain a low-frequency structural image and three high-frequency texture images;

Step 3: Perform bilateral texture filtering on the low frequency sub-band after wavelet transform to filter out texture details in the image and keep the image edges unambiguous;

Step 4: Perform multi-scale adaptive enhancement on each high frequency sub-band after wavelet transform to enhance texture detail information in the seismic image;

Step 5: performing inverse wavelet transform on the high and low frequency sub-bands processed in steps 3 and 4 to synthesize a new luminance component $Y'$;

Step 6: Perform contrast adaptive enhancement on $Y'$ to obtain the new luminance component $y$;

Step 7: Combine the luminance component $y$ with the chrominance components $Cb$ and $Cr$, and convert it into the RGB space, that is, obtain a seismic section image with enhanced texture details.

The following is an introduction to the implementation of important modules in the algorithm.

### 2.1 Wavelet decomposition of luminance component Y

The two-dimensional discrete wavelet transform includes two processes of decomposition and reconstruction, wherein the two-dimensional wavelet decomposition process is as follows: First, a one-dimensional wavelet transform is performed on the row data of the image to obtain the low-frequency component L and the high-frequency component H in the horizontal direction. Then, one-dimensional wavelet transform is performed on each column on the low and high-frequency components after the transformation, and four components LL, LH, HL, and HH are obtained in the horizontal and vertical directions, respectively. The reconstruction process of two-dimensional wavelet is as follows: first, one-dimensional inverse wavelet transform is performed on the image column data in the vertical direction, and then one-dimensional inverse wavelet transform is performed on the image row data in the horizontal direction, and finally the reconstructed image is obtained. The decomposition and reconstruction process are shown in Fig. 2.

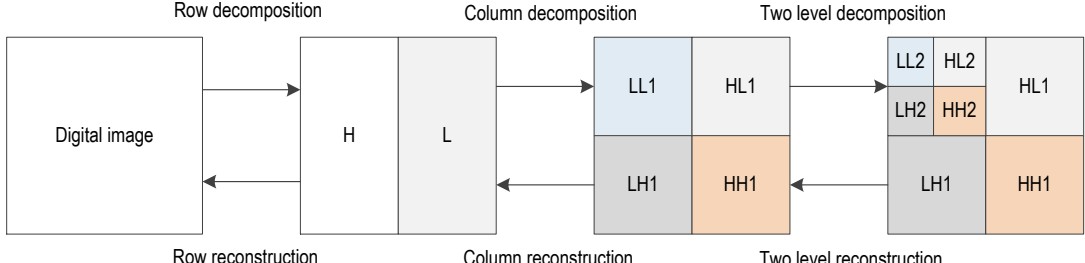

**Fig. 2 2-D discrete wavelet decomposition and reconstruction of image**

Considering the real-time performance of the calculation, the Haar wavelet is used to perform single-layer wavelet decomposition on the luminance component Y to obtain a low-frequency structural image and three high-frequency texture images. It can be seen from Fig. 2 that the obtained low frequency sub-band has an area of only 1/4 of the original image, so the bilateral texture filtering on the low frequency sub-band greatly improves the processing speed.

### 2.2 Bilateral texture filtering of wavelet low frequency subband

Before introducing bilateral texture filtering, review the bilateral filtering with groundbreaking work (Tomasi and Manduchi, 1998). Given an input image $I$, a reference image $G$, and the output of the bilateral filtering is $J$, the following relationship exists:

$$J_p = \frac{1}{k_p} \sum_{q \in \Omega_p} f\left(\|q - p\|\right) g\left(\|G_q - G_p\|\right) I_q, \qquad (1)$$

Where $k_p$ is the normalization factor, $J_p$ is the weighted mean of all $I_q$ of pixel $p$ in neighborhood $\Omega_p$, and f and g are two typical Gaussian functions.

The selection of the reference image $G$ has a great influence on the bilateral filtering effect, and the bilateral texture filtering is realized by cleverly selecting the reference image $G$; Bilateral texture filtering can better filter out texture details in images while keeping edges unambiguous (Cho et al., 2014). The core idea is to obtain the texture features by the method of partial block transfer, which can effectively realize the soft segmentation of the image texture region and preserve the structure of the image.

Assuming an image block of size $k \times k$, in image $I$, for each pixel p, a total of $k^2$ image blocks contain p, and an image block centered at q is $\Omega_q$. In these blocks, $\Omega_q$ is assumed to be a block containing the least significant structure edges. Once $\Omega_q$ is found to satisfy this property, we perform average filtering within this block to get the pixel value of the current point, denoted as $B_q$, and use this result as a bilateral Filtered reference image. Given the input image $I$, first apply the mean filter kernel of $k \times k$ to calculate the mean of the image $I$, denoted as $B$. For each pixel $p$, calculate its tonal range according to formula (2):

$$\Delta\left(\Omega_q\right) = I_{\max}\left(\Omega_q\right) - I_{\min}\left(\Omega_q\right), \qquad (2)$$





Where $I_{\max}\left(\Omega_q\right)$ and $I_{\min}\left(\Omega_q\right)$ represent the maximum and minimum values of the pixel values in the block, respectively; in the neighborhood of this pixel, the block with the smallest $\Delta\left(\Omega_q\right)$ is found, and the reference image $G_p$ in formula (1) is replaced with $B_q$.

Since the definition of the tonality range of formula (2) is relatively simple, the concept of related total variation is introduced to improve (Xu et al., 2012), and the correlation total variation is defined as:

$$\mathrm{mRTV}\left(\Omega_q\right)=\Delta\left(\Omega_q\right)\frac{\max\limits_{r\in\Omega_q}\left|\left(\partial I\right)_r\right|}{\sum\limits_{r\in\Omega_q}\left|\left(\partial I\right)_r\right|+\varepsilon}, \tag{3}$$

Where $\left|\left(\partial I\right)_r\right|$ denotes the gradient energy of $r\in\Omega_q$ and $\varepsilon$ is a small normal number, preventing the denominator from being zero.

From formula (3), the mRTV value will be very small in the smooth region of the image and also very sensitive to noise. To solve this problem, when assigning the $B_q$ value to the reference image $G_p$, it is necessary to check the mRTV values of $\Omega_p$ and $\Omega_q$, and if and only if $\mathrm{mRTV}\left(\Omega_q\right)<\mathrm{mRTV}\left(\Omega_p\right)$, assign the $B_q$ value to the reference image $G_p$, when the two mRTV values are close , assigning $B_p$ value to the reference image $G_p$; To achieve this idea, the final reference image $G'$ is obtained by linear interpolation between images $B$ and $G$:

$$G'_p=\alpha_p G_p+\left(1-\alpha_p\right)B_p, \tag{4}$$

where

$$\alpha_p=2\left(\frac{1}{1+\exp\left(-\sigma_\alpha\left(\mathrm{mRTV}\left(\Omega_p\right)-\mathrm{mRTV}\left(\Omega_q\right)\right)\right)}-0.5\right). \tag{5}$$

In the formula, the weight $\alpha_p\in\left[0,1\right]$ is small in the smooth and textured regions, but larger in the vicinity of the boundary; $\sigma_\alpha$ controls the degree of change from the edge to the smooth/texture transition, generally taking $\sigma_\alpha=5k$.

The bilateral texture filtering steps are summarized as follows:

Step 1: Input image $I$, average filtering image $I$ to obtain $B$;

Step 2: Calculate the mRTV value of each pixel $p$ in the image $I$ according to the formula (3);

Step 3: For each pixel $p$, find the pixel $q$ with the smallest mRTV value in $\Omega_p$, let $G_p=B_q$;

Step 4: Calculate the reference image $G'$ using formula (4);

Step 5: Using $G'$ as the reference image and $I$ as the input image into formula (1), the bilateral texture filtering of image $I$ can be realized, and the filtered image $J$ is output.

### 2.3 Detail enhancement of wavelet high frequency subband

In order to enhance the useful information of the texture while suppressing the noise information, the wavelet high frequency sub-band is detail enhanced, That is, the following adaptive enhancement transform is used for each





high-frequency sub-band obtained by wavelet transform (Zhang et al., 2006). The specific transformation formula is as follows:

$$f(x) = \alpha \left[ sigm(c(x-b)) - sigm(-c(x+b)) \right], \qquad (6)$$

where

$$\begin{cases} \alpha = \dfrac{1}{sigm(c(1-b) - sigm(-c(1+b)))}; \\ 0 < b < 1; \\ sigm(x) = \dfrac{1}{1+e^{-x}}; \end{cases} \qquad (7)$$

b and c are used to control the magnitude of the enhancement. As can be seen from Fig. 3, the function $f(x)$ mainly enhances the middle portion of the image gray value, the smaller coefficient corresponds to noise, and the larger coefficient corresponds to the texture detail of the image.

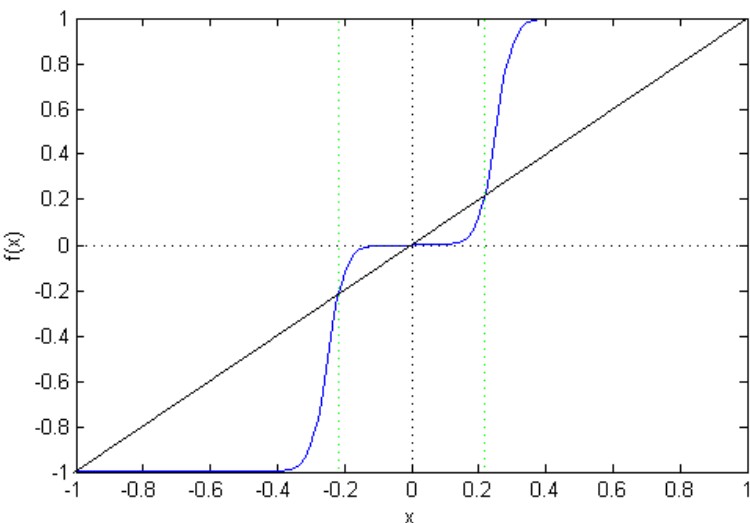

**Fig. 3 The graph of adaptive enhancement transform function, in which $b = 0.25$, $c = 40$**

After processing the low frequency sub-band and the high frequency sub-band according to the method given in Sections 2.2 and 2.3, the inverse wavelet transform can be used to obtain the luminance component *Y* after the texture detail is enhanced; By synthesizing the luminance component *Y* and the chrominance components *Cb* and *Cr* into an RGB format, a seismic section image with enhanced texture details can be obtained.

**3. Experimental results and analysis**

In order to verify the effectiveness of the proposed algorithm, we selected two representative enhancement algorithms, wavelet enhancement (Sakellaropoulos et al., 2003) and bilateral filtering enhancement (Farbman et al., 2008).


Experiment 1: Fig. 4(a) is a seismic section image. Due to seismic data noise and noise introduced during seismic data processing, the image texture information is not clear, which affects subsequent image interpretation. According to our algorithm idea, first, Fig. 4(a) is converted from the RGB format to the *YCbCr* format, and the luminance component *Y* and the chrominance components *Cb* and *Cr* are obtained as shown in Fig. 4(b)(c)(d). Single-layer Haar

wavelet decomposition is performed on the luminance component Y to obtain a low-frequency structural component *cA* and three high-frequency detail components *cH*, *cV* and *cD*, as shown in Fig. 5; Performing bilateral texture filtering on the component *cA* shown in Fig. 5(a), wherein the block size is k=3, and the filtered result is as shown in Fig. 6(a); Detail enhanced for the components cH, cV, and *cD* shown in Fig. 5(b)(c)(d), and the enhanced results are shown in Fig. 6(b)(c)(d). Perform wavelet reconstruction on the component shown in figure 6 to form the brightness component *Y*, and

then convert it into RGB format after combining with the chromaticity component *CbCr*., thereby obtaining a seismic section image with enhanced texture details, as shown in Fig. 7(c). The enhanced results using wavelet transform and bilateral filtering are shown in Fig. 7(b)(c).

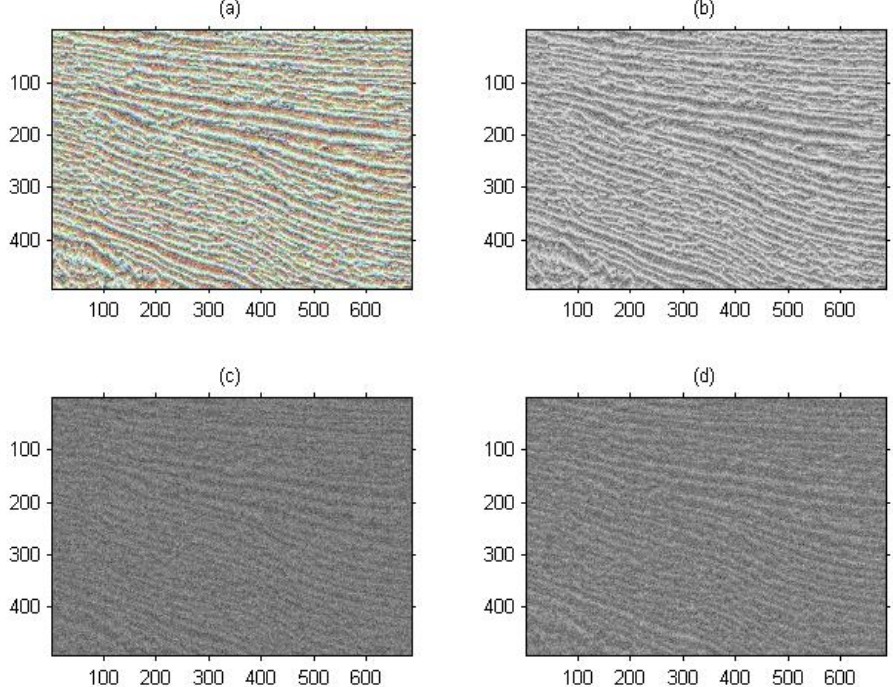

**Fig. 4 Converting a seismic section image from RGB format to YcbCr format; (a) Original seismic image; (b)**

**The brightness component *Y* of the image (a) after the format conversion; (c), (d) Chroma component *Cb* and *Cr* of image (a) after format conversion**



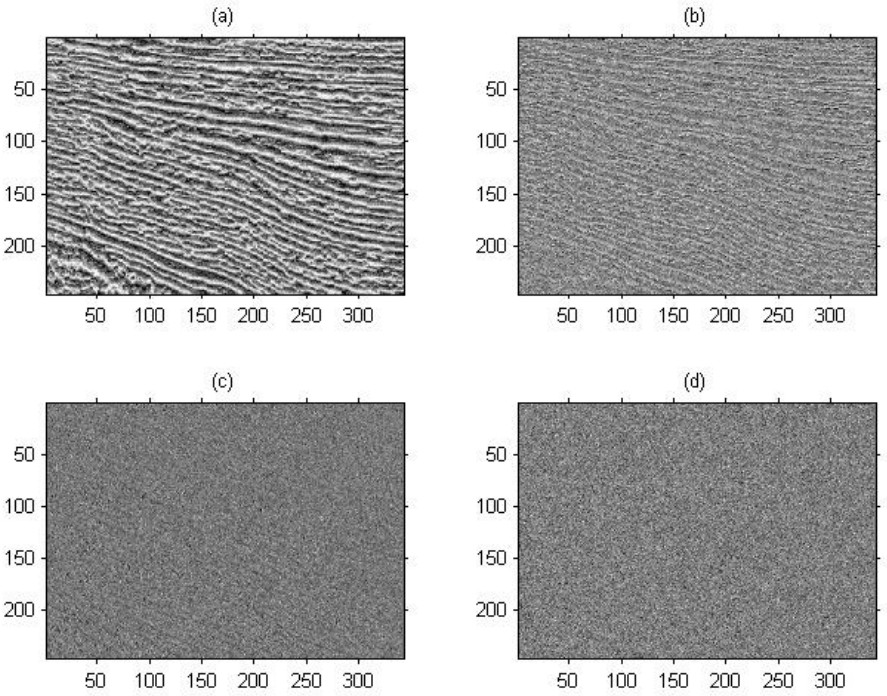

**Fig. 5 The result of 2-D discrete wavelet decomposition of the luminance component $Y$; (a) Low frequency structure image component $cA$; (b) Lateral high frequency detail component $cH$; (c) Longitudinal high frequency detail component $cV$; (d) Diagonal high frequency detail component $cD$.**

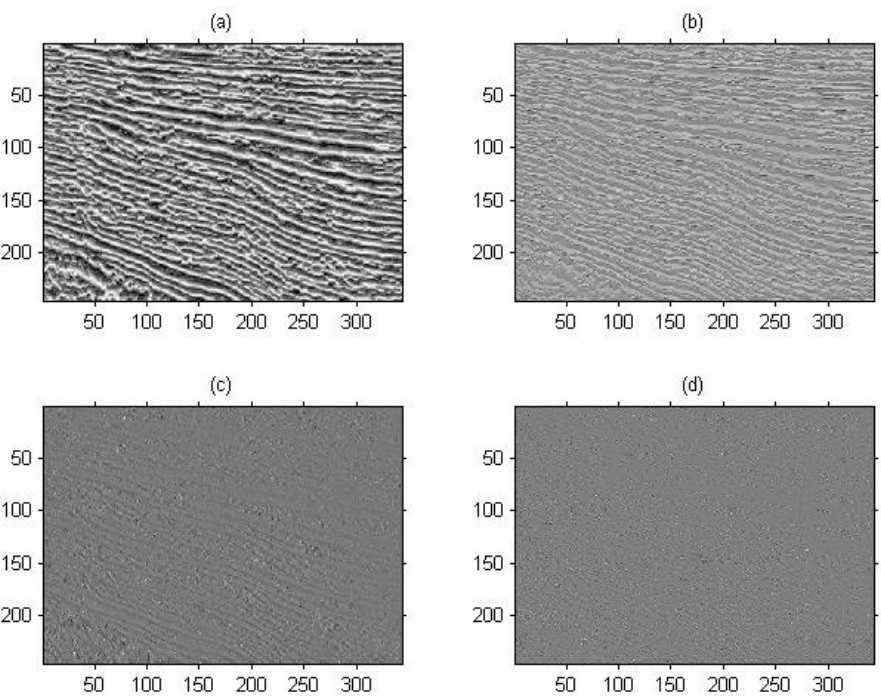



**Fig. 6 The result of the enhancement processing for each component in Fig. 5; (a) The results obtained by bilateral texture filtering of Fig 5(a); (b) Obtaining the results after detail enhancement of Fig 5(b); (c) Obtaining the results after detail enhancement of Fig 5(c); (d) The results are enhanced by the detail enhancement of Fig 5(d).**

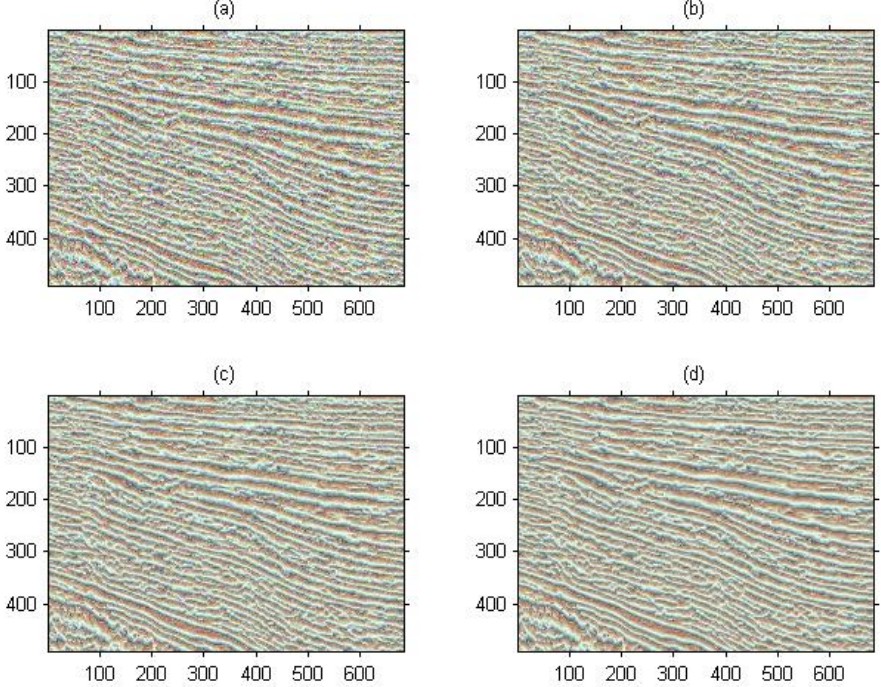

**Fig. 7 Comparison of algorithm results; (a) Original seismic section image; (b) Wavelet enhancement algorithm; (c) Bilateral filtering enhancement algorithm; (d) our algorithm.**

As can be seen from Fig.7: The wavelet transform enhancement method is mainly for the enhancement of point singular features, and there is often no good processing result for the lines in the image. Therefore, the large-area

oscillation period texture in the seismic section image cannot be well processed, and a false edge is generated during the processing; Although the bilateral filtering enhancement method can enhance the image edge information well, the enhancement effect of the texture part is still unsatisfactory; The method we proposed gives full play to the advantages of bilateral texture filtering in image texture processing, which can better enhance the unclear texture details in the original image, and at the same time enhance the texture information without destroying the image edge information,

thus obtaining the rich texture details of the seismic section image. It not only enhances the visualization effect, but also provides more abundant and accurate information for the geologist's next seismic section image interpretation.

Experiment 2: Fig.8 shows the comparison results of the second set of experimental image enhancements, where Fig. 8(a) is the original image. From the visual point of view, the method of this paper better enhances the unclear details in the original image and obtains rich texture features. Fig. 8(b) is a wavelet enhancement result. This method

achieves a better enhancement effect in the region where the point information is rich in the original image, but the enhancement effect on the linear texture information is not obvious; Fig 8(c) shows the effect of bilateral filtering enhancement. This method has a large advantage in smaller wave enhancement, but the overall detail enhancement



effect is far from Fig. 8(d). Our method not only improves the contrast of the image, but also has a good enhancement effect on the edge of the seismic section image and texture details.

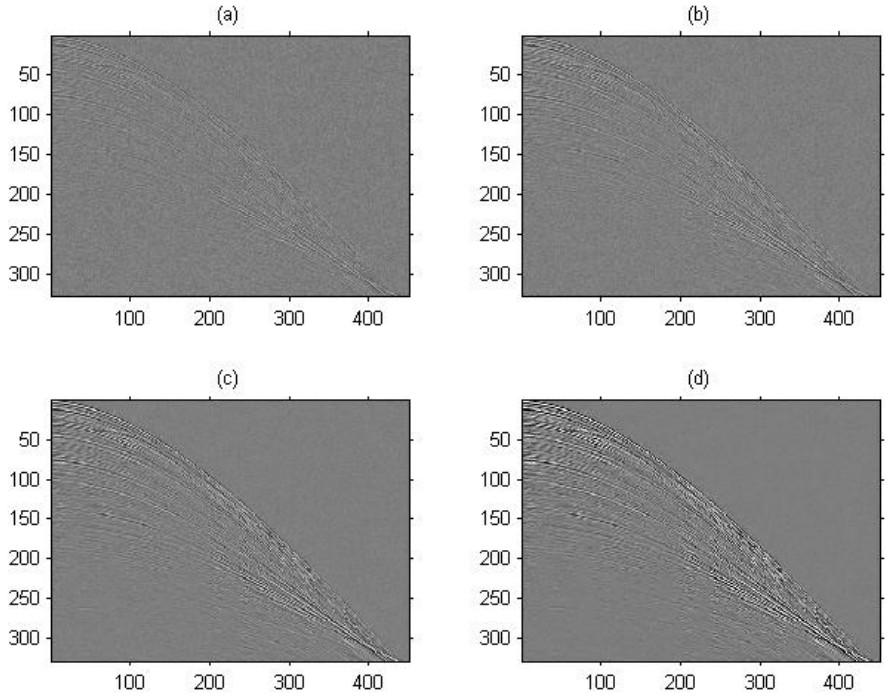

**Fig. 8 The results of our algorithm and comparison algorithm. (a) original seismic section images; (b) wavelet**
5    **enhancement algorithm; (c) bilateral filtering enhancement algorithm; (d) our algorithm.**

Experiment 3: this experiment mainly tests the processing time of the algorithm in this paper. The CPU of the test computer was Intel(R) Core (TM) i5-4590 3.30ghz, the RAM size was 4G, and the software platform was MATLAB. The running time of the proposed algorithm is several times slower than that of the bilateral filtering enhancement algorithm, but the enhancement effect is better. Table 1 shows the time-consuming statistics of the various components
10    of the algorithm in the test platform. The picture used in the test is a grayscale image of 800*600.

**Table 1 Time consumption of each part of the algorithm in this paper (/s)**

| No | Component | $k=3$ | $k=5$ | $k=7$ |
|----|-----------|-------|-------|-------|
| 1 | DWT | 0.1172 | 0.1172 | 0.1172 |
| 2 | Computing mRTV | 0.4847 | 0.9160 | 1.3031 |
| 3 | Computing $B$ | 0.0105 | 0.0105 | 0.0105 |
| 4 | Patch shift | 0.3894 | 0.9072 | 1.1622 |
| 5 | Computing $\alpha$ | 0.0166 | 0.0166 | 0.0166 |
| 6 | Computing Eq. (1) | 0.1546 | 0.5125 | 0.7147 |
| 7 | Computing Eq. (6) | 0.0272 | 0.0272 | 0.0272 |
| 8 | i-DWT | 0.0670 | 0.0670 | 0.0670 |
| 9 | Total | 1.2672 | 2.5742 | 3.4185 |



## 4. Conclusions

We proposed a texture detail enhancement method for seismic section image. Wavelet transform can effectively separate structure information (low frequency subband) and detail information (high frequency subband) of image. High frequency noise in structural information can be estimated and removed effectively by using bilateral texture filter in low frequency subband. In the high frequency subband, adaptive enhancement transform can be used to enhance the image edge and texture information, and effectively remove the low frequency noise. Experimental results show that the algorithm can not only effectively enhance the edge and texture information of the image, but also reduce the impact of noise, and the algorithm has better real-time performance.

*Data availability.* Data and results of calculations are available by e-mail request.

*Author contributions.* XYJ designed the study, performed the research, analyzed data, and worte the paper. CLDY contributed to refining the ideas, carrying out additional analyses, and finalizing this paper.

*Competing interests.* The authors declare that they have no conflict of interest.

*Financial support.* This work was supported by the Natural Science Foundation of Shandong Province, China (ZR2018MEE008), the Key Research and Development Program of Shandong Province, China (2017GSF20115).

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
