# Peer review of "Seismic section image detail enhancement method based on bilateral texture filtering and adaptive enhancement of texture details"

_Nonlinear Processes in Geophysics, 2019_

## Referee Comment (RC1) · Sergio Chávez-Pérez (Referee) · 10 Nov 2019

Your work is interesting and potentially useful. However, I think you need to improve how you put the obtained results into context, with relevant references from the exploration geophysics (seismology) community, as the topic is not necessarily new for such group. Did I explain myself?

Highlighting the implications of your work for the exploration seismology community will help your efforts. Otherwise, it will only sound like another wavelet transform paper working on a rather old topic.

Hope this helps!

Yours sincerely,

[Figure]

Sergio Chávez-Pérez Mexican Petroleum Institute Mexico City

---

## Referee Comment (RC2) · Anonymous Referee #2 · 20 Jan 2020

Dear Editor and dear authors, First of all I apologize for the delay in finalizing the review. GENERAL COMMENTS: The manuscript proposes a new method for the enhancement of some important details for seismic section images. In the introduction, a complete overview of the state of the art of the existing methodologies for the image processing is provided, highlighting the limits of the different categories of methods. Then, the new method is described, and an its application to two different examples of seismic sections is shown. Through them, the improvement in the quality of seismic sections is shown with respect to the original image and the ones provided by other methods, mainly in the second application. The presentation of the manuscript is clear and coincise; the text, in the most part, is fluent and precise; the figures are of good quality. My only substantial request to the authors is to provide further details about

how these improvements in the image processing can help people to read seismic sections. In particular, referred to the examples provided in the manuscript, the better resolution achieved with your method is able, for example, to highlight some useful details, for geological/exploration purposes, that were completely hidden in the original image or in the ones retrieved by other techniques? On these grounds, my suggestion is to accept the manuscript after a very minor revision. TECHNICAL COMMENTS: Page 1, line 22: "It is of great significance" ... Can you specify with more detail what is of great significance? Page 1,line 30: "etc;" change into "etc." Page 1, line 32; "The" change into "the". Concerning this, there are many similar typos along the manuscript, and also in the abstract, i.e. capital letter after semicolomns. Please fix it. Page 2,line 2: "Then using a high-pass ..." Reformulate into "Then a high pass filter in the Fourier transform domain is used" Page 2, line 4: Can you better specify what do you mean with high and dark areas? Page 2, lines 7-10: "Such methods perform ..." Please reformulate. Page 2, line 12: Can you say what do circular effects on the edge consist in? Page 2, line 25: "The experimental ..." I suggest to move this sentence to the conclusions or to directly remove it. Page 2, line 29: "In this paper, the ..." change into "The wavelet ..." removing "In this paper". Page 3, Step 5: Could you please insert the symbol Y' in the figure 1, as described in the step 5? Page 3, Fig.1 Please indicate the "contrast adaptive enhancement" described in the step 6 in figure 1. Page 4, line 13: what is q? Please introduce it here. Page 4 , line 19-20: "Assuming an image block ...". Can you reformulate in a simpler way this sentence? It is too rich in asides. Page 7, line 1 and following ones: I suggest to write the different passages as a list; Page 7, line 6-7: "Performing ..."I suggest to reformulate this sentence. Page 7, line 8: "Detail enhanced ...." Do you mean "Detail enhancement?". Please reformulate the sentence Page 7, line 11: Fig 7 (c) $\longrightarrow$ May be Fig. 7(d)? General comment about the presentation of the figures 5 and 6: I suggest to rearrange them so that the reader could directly see the effect of the processing on the different components of the image. For example, you show in figure 5a the low frequency image component; in figure 6a the results of the bilateral texture filtering on figure 5a. In my opinion it could be

more useful to show the images before and after the processing in a single figure. The same suggestion is for the other components of the image (cH, cV and cD). Page 9, line 8 and following ones: In the description of the final images, retrieved with different algorithms, it could be very useful to refer to specific points that you could mark on the images: For example, you write : "The large-area oscillation period texture in the seismic section image cannot be well processed and a false edge is generated during the processing". In my opinion, these features should be marked on the images, so that the reader could better understand the improvements in the image resolution retrieved by your algorithm. Page 9, line 20 and 21: can you explain the differences between point information and linear texture information? Page 10, table 1: Could you please better describe this table, reminding the meaning of k ? Page 11, line 9; "better real-time performance". May be I misunderstood, but in the previous page you wrote "The running time of the proposed algorithm is several times slower than that of the bilateral filtering"

---

## Author Comment (AC1) · 6 Feb 2020

Response to Comments

Ref: npg-2019-46 Title: Seismic section image detail enhancement method based on wavelet transform Corresponding author: Dr. Chang-Lei DongYe E-mail: dycl.cn@163.com

Dear Editors and Reviewers, On behalf of my co-authors, we thank you very much for giving us an opportunity to revise our manuscript; we appreciate editor and reviewers very much for their positive and constructive comments and suggestions on our manuscript. We have studied comments carefully and have made correction which we hope meet with approval. Attached please find the revised version, which we would like

to submit for your kind consideration. We would like to express our great appreciation to you and reviewers for comments on our paper. Looking forward to hearing from you. Thank you and best regardsïijĄ

Yours sincerely, Chang-Lei DongYe E-mail: dycl.cn@163.com

Referee #1 (Sergio Chávez-Pérez): Comments: Your work is interesting and potentially useful. However, I think you need to improve how you put the obtained results into context, with relevant references from the exploration geophysics (seismology) community, as the topic is not necessarily new for such group. Did I explain myself? Highlighting the implications of your work for the exploration seismology community will help your efforts. Otherwise, it will only sound like another wavelet transform paper working on a rather old topic. Author's response: Special thanks to you for your good comments. The goal of our paper is to improve the texture details of seismic section images by using image processing technology, so that seismologists can better understand the information in them. The wavelet decomposition used is only a means of image decomposition, but our paper topic focuses on wavelet decomposition, which is not appropriate. Author's changes in manuscript: We modified the manuscript title as: "Seismic section image detail enhancement method based on bilateral texture filtering and adaptive enhancement of texture details". So that it can highlight the theme of the manuscript. In addition, we also revised section 1 of the paper and added relevant references according to your suggestion. For more details, please see Supplement PDF P2, L18 - 27

Referee #2: General comments: The manuscript proposes a new method for the enhancement of some important details for seismic section images. In the introduction, a complete overview of the state of the art of the existing methodologies for the image processing is provided, highlighting the limits of the different categories of methods. Then, the new method is described, and an its application to two different examples of seismic sections is shown. Through them, the improvement in the quality of seismic sections is shown with respect to the original image and the ones provided by other

methods, mainly in the second application. The presentation of the manuscript is clear and concise; the text, in the most part, is fluent and precise; the figures are of good quality. My only substantial request to the authors is to provide further details about how these improvements in the image processing can help people to read seismic sections. In particular, referred to the examples provided in the manuscript, the better resolution achieved with your method is able, for example, to highlight some useful details, for geological/exploration purposes, that were completely hidden in the original image or in the ones retrieved by other techniques? On these grounds, my suggestion is to accept the manuscript after a very minor revision. Author's response: It is really true as Reviewer suggested that we should provide further details about how these improvements in the image processing can help people to read seismic sections. Author's changes in manuscript: We have added a discussion on the above issues, and the red box is marked in Fig. 7, for more details, please see Supplement PDF P15, L1 - 7, P14, Fig. 7.

Technical comments: Q1: Page 1, line 22: "It is of great significance" ... Can you specify with more detail what is of great significance? A1: We have re-written this part according to the Reviewer's comments. See Supplement PDF P1, L22 - 25.

Q2: Page 1, line 30: "etc;" change into "etc." A2: We have changed it. See P1, L33.

Q3: Page 1, line 32; "The" change into "the". Concerning this, there are many similar typos along the manuscript, and also in the abstract, i.e. capital letter after semicolomns. Please fix it. A3: We have fixed it according to Reviewer's comments.

Q4: Page 2, line 2: "Then using a high-pass ..." Reformulate into "Then a high pass filter in the Fourier transform domain is used" A4: We have reformulated it. See P2, L5 - 7.

Q5: Page 2, line 4: Can you better specify what do you mean with high and dark areas? A5: We have re-written this part. See P2, L8.

Q6: Page 2, lines 7-10: "Such methods perform ..." Please reformulate. A6: We have re-written this sentence. See P2, L10 -11.

Q7: Page 2, line 12: Can you say what do circular effects on the edge consist in? A7: We have re-written this sentence. See P2, L17.

Q8: Page 2, line 25: "The experimental ..." I suggest to move this sentence to the conclusions or to directly remove it. A8: We have removed this sentence. See P2, L37 - 39.

Q9: Page 2, line 29: "In this paper, the ..." change into "The wavelet ..." removing "In this paper". A9: We have changed it. See P3, L2.

Q10: Page 3, Step 5: Could you please insert the symbol Y' in the figure 1, as described in the step 5? Page 3, Fig.1 Please indicate the "contrast adaptive enhancement" described in the step 6 in figure 1. A10: We have revised Fig. 1 according to Reviewer's suggestion. See P3, Fig. 1.

Q11: Page 4, line 13: what is q? Please introduce it here. A11: We have added the meaning of q. See P4, L28.

Q12: Page 4 , line 19-20: "Assuming an image block ...". Can you reformulate in a simpler way this sentence? It is too rich in asides. A12: We have reformulated this part according to your suggestion. See P5, L6 - 8.

Q13: Page 7, line 1 and following ones: I suggest to write the different passages as a list; A13: We have written the part as a list according to Reviewer's suggestion. See P7, L13 - 24.

Q14: Page 7, line 6-7: "Performing ..."I suggest to reformulate this sentence. A14: We have reformulated this sentence. See P7, L19 - 21.

Q15: Page 7, line 8: "Detail enhanced ...." Do you mean "Detail enhancement?". Please reformulate the sentence A15: We have reformulated this sentence. See P7,

L21 - 22.

Q16: Page 7, line 11: Fig 7 (c) −→ May be Fig. 7(d)? A16: We are very sorry for our incorrect writing. See P8, L1.

Q17: General comment about the presentation of the figures 5 and 6: I suggest to rearrange them so that the reader could directly see the effect of the processing on the different components of the image. For example, you show in figure 5a the low frequency image component; in figure 6a the results of the bilateral texture filtering on figure 5a. In my opinion it could be more useful to show the images before and after the processing in a single figure. The same suggestion is for the other components of the image (cH, cV and cD). A17: We have rearranged Fig. 5 and Fig. 6 according to the Reviewer's suggestion. See P10, Fig. 5.

Q18: Page 9, line 8 and following ones: In the description of the final images, re-trieved with different algorithms, it could be very useful to refer to specific points that you could mark on the images: For example, you write : "The large-area oscillation pe-riod texture in the seismic section image cannot be well processed and a false edge is generated during the processing". In my opinion, these features should be marked on the images, so that the reader could better understand the improvements in the image resolution retrieved by your algorithm. A18: We have re-written this part according to the Reviewer's suggestion. See P12, Fig. 6.

Q19: Page 9, line 20 and 21: can you explain the differences between point information and linear texture information? A19: We have rewritten this part to make it clearer. See P13, L12 - 14.

Q20: Page 10, table 1: Could you please better describe this table, reminding the meaning of k ? A20: We have re-written this part according to the Reviewer's sugges-tion. See P15, L10 - 16.

Q21: Page 11, line 9; "better real-time performance". May be I misunderstood, but in

Interactive
comment

the previous page you wrote "The running time of the proposed algorithm is several times slower than that of the bilateral filtering" A21: We are very sorry for our inaccurate writing, we have deleted "better real-time performance". See P15, L26.

Please also note the supplement to this comment:
https://www.nonlin-processes-geophys-discuss.net/npg-2019-46/npg-2019-46-AC1-supplement.pdf

---

## Author Response (AR2)

**Response to Comments**

Ref: npg-2019-46

Title: Seismic section image detail enhancement method based on wavelet transform

Corresponding author: Dr. Chang-Lei DongYe

E-mail: dycl.cn@163.com

Dear Editors and Reviewers,

On behalf of my co-authors, we thank you very much for giving us an opportunity to revise our manuscript; we appreciate editor and reviewers very much for their positive and constructive comments and suggestions on our manuscript.

We have studied comments carefully and have made correction which we hope meet with approval. Attached please find the revised version, which we would like to submit for your kind consideration.

We would like to express our great appreciation to you and reviewers for comments on our paper. Looking forward to hearing from you.

Thank you and best regards!

Yours sincerely,

Chang-Lei DongYe

E-mail: dycl.cn@163.com

**Referee #2 :**

**Comments:** I really appreciated the effort done by you to improve the already good quality of your manuscript.

Nonetheless, I will point out some (just five) technical corrections which could further increase the precision of the manuscript:

- Page 1, line 23: Change "plays" into "play".

- Page 5, line 7: I still do not agree with the syntax of this sentence, in particular when you say

"marked an image block centered at q ..."

- Page 7, line 19: Change "show" into "shown".

- Page 7, line 21: Change "used" into "applied".

- Page 12 and 13: In my opinion, it is important to explain what is the red box both in the caption and in the main text. In particular, when you describe the results of the processing of the image you can refer to the red box in the figure. The same, in my opinion, has to be done for figure 7.

**Author's response:** We have made correction according to the Reviewer's comments. For details, please refer to supplement as attachment.

We tried our best to improve the manuscript and made some changes in the manuscript. These changes will not influence the content and framework of the paper. And here we did not list the changes but marked in revised paper.

We appreciate for Editors and Reviewers' warm work earnestly, and hope that the correction will meet with approval.

Once again, thank you very much for your comments and suggestions.